# Interleukin-6 from *Mycobacterium abscessus*-infected macrophages enhances the survival of B cell-derived plasmablasts *in vitro*

Issara Prachongsai,[1,2] Rawipas Saisuwan,[3] Pattanan Smosorn,[1,3] Nuttawan Kassaket,[3] Wiwit Tantibhedhyangkul[3]

**ABSTRACT** *Mycobacterium abscessus* (MABS), a rapidly growing non-tuberculous mycobacterium, causes disseminated infections in patients with adult-onset immunodeficiency due to anti-IFN-γ autoantibodies. We investigated whether cytokines produced by MABS-infected cells contribute to plasmablast survival and antibody production. To address this, we first analyzed the cytokine response in monocyte-derived macrophages infected with MABS and found that IL-6 was secreted at high levels. Since IL-6 is known to be produced by fibroblasts, we next characterized the intracellular survival of MABS and its ability to induce cytokine production in fibroblasts. We demonstrated that MABS invaded fibroblasts, replicated intracellularly, and induced high levels of IL-6 secretion. Given the established role of IL-6 in promoting plasmablast survival, we next developed an *in vitro* culture of B cell-derived plasmablasts to test the effect of IL-6. To do this, B cells from healthy donors were differentiated into early plasmablasts using Toll-like receptor (TLR) 7/8 agonist. After 4 days, plasmablasts were then cultured for an additional 3 days with either recombinant IL-6 or supernatants from infected macrophages. Our results demonstrated that both recombinant IL-6 and supernatants from infected macrophages promoted plasmablast survival and expansion. Blocking the IL-6 receptor with tocilizumab, but not inhibiting JAK1/2 with baricitinib, reduced plasmablast survival and IgG secretion in cultures treated with infected macrophage supernatants. Collectively, these findings suggest that infection-induced IL-6 promotes plasmablast survival and antibody production. Targeting IL-6 signaling could, therefore, represent a potential therapeutic strategy to modulate antibody responses.

**IMPORTANCE** The difficult-to-treat MABS infection is a major clinical problem in Asian patients with adult-onset primary immunodeficiency associated with anti-IFN-γ autoantibodies. Understanding the cytokine responses induced by MABS may support the development of cytokine-targeted therapies that control autoantibody production. In this study, we found that IL-6 was produced at high levels by both macrophages and fibroblasts following infection. Moreover, we demonstrated that IL-6 is a key cytokine promoting plasmablast survival, as IL-6 receptor blockade with tocilizumab significantly reduced plasmablast viability in cultures stimulated with infected macrophage supernatants. Together, these findings provide a rationale for future clinical investigation of cytokine-targeted therapeutic approaches.

**KEYWORDS** *Mycobacterium abscessus*, interleukin-6, macrophages, fibroblasts, plasmablast, tocilizumab, anti-IFN-γ autoantibody

**Peer Reviewers** Peng Gao, University of Hong Kong, Hong Kong, Hong Kong; Bethany Patenall, University of Bath, Bath, United Kingdom

Address correspondence to Wiwit Tantibhedhyangkul, wiwit.tan@mahidol.ac.th.

The authors declare no conflict of interest.

**M**ycobacteroides abscessus (formerly *Mycobacterium abscessus*, MABS) is a rapidly growing non-tuberculous mycobacterium that typically causes skin and soft tissue infections, as well as lung infections in patients with underlying chronic lung

diseases such as bronchiectasis and cystic fibrosis (1, 2). Beyond organ-specific infections, this pathogen can also cause disseminated infections in patients with adult-onset primary immunodeficiency (AOID) caused by anti-interferon gamma (IFN-γ) autoantibody production (3). MABS infection is one of the most difficult-to-treat mycobacterial infections due to its acquired resistance to multiple classes of antibiotics. Based on colony morphology, MABS can be classified into two variants: smooth and rough colonies. The rough variant lacks glycopeptidolipids in the cell wall and is more virulent compared to the smooth variant (1, 2).

AOID associated with anti-IFN-γ autoantibodies is a primary immunodeficiency predominantly found in individuals from Southeast and East Asian countries. These patients typically have high levels of neutralizing anti-IFN-γ autoantibodies. Since IFN-γ is a critical cytokine for the type 1 immune response, these patients frequently experience recurrent opportunistic infections caused by intracellular pathogens, including non-tuberculous mycobacteria (particularly rapidly growing mycobacteria and *M. avium* complex), *Talaromyces marneffei*, *Cryptococcus*, non-typhoidal *Salmonella*, and varicella-zoster virus (4–6). The pathogenic mechanisms underlying the production of anti-IFN-γ autoantibodies remain unclear. However, evidence suggests that both genetic and environmental factors play a role. Specific HLA class II alleles, namely, HLA-DRB1*15:02-DQB1*05:01 and HLA-DRB1*16:02-DQB1*05:02, have been reported to be associated with the disease (7–9). A previous study identified a B-cell linear epitope in the C-terminal region of the IFN-γ protein, with an amino acid sequence highly similar to that of the Noc2 protein from *Aspergillus* spp. Furthermore, patient plasma containing anti-IFN-γ autoantibodies was found to cross-react with the Noc2 peptide. This evidence suggests that exposure to this *Aspergillus* protein may act as a trigger for the production of these autoantibodies (10).

Since IFN-γ is a protein antigen, the production of anti-IFN-γ autoantibodies is likely mediated by T cell-dependent B cell activation. However, several cytokines and growth factors produced by innate immune cells—such as IL-6, IL-10, type I IFNs, B cell-activating factor (BAFF), and a proliferation-inducing ligand (APRIL)—can also promote B cell activation, enhance antibody responses, and support plasma cell survival (11–14). Indeed, infection is a well-recognized factor that can exacerbate or trigger the relapse of autoimmune diseases (15). It is well established that cytokines produced during infection enhance B cell activation and plasma cell differentiation, leading to increased antibody production. For example, viral infections stimulate plasmacytoid dendritic cells to secrete IL-6 and type I interferons, which, in turn, promote the differentiation of B cells into antibody-producing plasma cells (16). In animal models, myeloid dendritic cells and monocytes/macrophages have been shown to produce IL-6, which supports plasma cell survival within lymph nodes (17). Additionally, macrophages can induce plasma cell differentiation through an IP-10 (CXCL10)-dependent mechanism (18). Moreover, in AOID patients, levels of anti-IFN-γ autoantibodies are increased during active infections. Based on these findings, we hypothesize that specific cytokines induced during active opportunistic infections in patients with this immunodeficiency disease contribute to the persistent production of anti-IFN-γ autoantibodies.

Several immunosuppressive agents have been tried to reduce anti-IFN-γ autoantibody levels, as decreases in these autoantibodies are associated with improved clinical outcomes of infection. Commonly used therapies include B cell-depleting agents (anti-CD20 monoclonal antibodies, such as rituximab) as well as non-specific immunosuppressive agents, including corticosteroids and cyclophosphamide. However, these immunosuppressive approaches have shown inconsistent or variable efficacy across studies (6, 19, 20). Moreover, non-specific immunosuppressive drugs adversely affect not only B cells but also T cells and phagocytes, potentially compromising host immunity to infection. Therefore, to identify novel therapeutic targets, we focus on macrophage-derived cytokines that may directly regulate antibody-secreting cells, including plasmablasts and plasma cells.

Given that *Mycobacterium abscessus* is the most common opportunistic infection in immunodeficient patients with anti-IFN-γ autoantibodies (4, 21), we selected this pathogen as a model to investigate the role of infection-induced cytokines in supporting the survival of B cell-derived plasmablasts. Additionally, due to its rapid replication, MABS is likely to induce a stronger inflammatory response compared to other pathogens. In this study, we demonstrate that high levels of IL-6 are produced by both macrophages and fibroblasts. We further confirm that IL-6 plays a key role in promoting plasmablast survival and provide additional support for the role of myeloid cell-derived innate cytokines in regulating antibody production.

## MATERIALS AND METHODS

### Bacterial culture

*Mycobacterium abscessus* strain ATCC 19977 (rough morphotype) was cultured in Middlebrook 7H9 broth supplemented with 0.05% Tween-80, 0.2% glycerol, and albumin-dextrose-catalase (ADC; Himedia, Mumbai, India) at 37°C. For cryopreservation, bacterial cultures were centrifuged at $3,000 \times g$ for 15 min. The bacterial pellet was resuspended in RPMI 1640 medium containing 20% fetal bovine serum (FBS) and 10% dimethyl sulfoxide (DMSO) and then passed repeatedly through a 26-gage needle to minimize clumping. The suspension was aliquoted and stored at –80°C until use.

### Cell culture: macrophages and fibroblasts

Peripheral blood mononuclear cells (PBMCs) were isolated from whole blood of healthy donors by density gradient centrifugation using PBMC separation medium (Pancoll human, density 1.077 g/mL; PAN-Biotech, Aidenbach, Germany). The PBMC isolation protocol was approved by the Institutional Review Board (COA Nos. Si 641/2019 and Si 509/2023). Monocytes were allowed to adhere to tissue culture plates for 90 min. Non-adherent lymphocytes were then removed, and adherent monocytes were cultured in RPMI 1640 supplemented with 10% human AB serum (Sigma-Aldrich, Saint Louis, MO), 10% fetal bovine serum (FBS; HyClone, Logan, UT), and granulocyte-macrophage colony-stimulating factor (GM-CSF) at a low dose (5 U or 0.5 ng/mL; ImmunoTools, Friesoythe, Germany). An equal volume of RPMI 1640 containing 10% FBS was added on day 2. From day 4 onward, monocyte-derived macrophages were maintained in RPMI 1640 with 10% FBS and used for infection experiments between days 7 and 9. Macrophage differentiation was confirmed by downregulation of CD14 and upregulation of surface CD68 expression (Data S1).

Human dermal fibroblasts (HDFs; ATCC PCS-201-012) were cultured in DMEM supplemented with 12% FBS, 1× non-essential amino acids (PAN-Biotech), and fibroblast growth factor (FGF-basic, 5 ng/mL; ImmunoTools). The culture medium was replaced every 2 days. Near-confluent monolayers were subcultured using 0.25% trypsin in 0.5 mM EDTA and seeded into tissue culture plates for infection experiments.

### Infection procedures

To assess cytokine responses, human monocyte-derived macrophages and human dermal fibroblasts were infected with MABS at a multiplicity of infection (MOI) of 20:1 because our preliminary data showed that infection at this MOI induced high levels of cytokine expression with minimal cell death (Data S2). Cell lysates were collected at 6 and 24 h post-infection for RNA extraction, while culture supernatants were harvested at 9 and 24 h post-infection for cytokine quantification by ELISA. Pam3Cys-Ser-(Lys)$_4$ 1 µg/mL (Abcam, Cambridge, United Kingdom), a TLR1/TLR2 agonist, was used as a positive control for macrophage stimulation because mycobacterial cell wall components are mainly recognized by TLR2 (22).

To obtain macrophage supernatants for B cell/plasmablast stimulation, macrophages were infected with MABS for 2 h. The bacterial inoculum was then gently removed, and

the infected macrophages were maintained in RPMI 1640 supplemented with 2% FBS and amikacin (20 mg/L) to inhibit the synthesis and secretion of bacterial proteins, which may be cytotoxic to B cells/plasmablasts. Under light microscopy, adherent MABS bacilli remained associated with macrophages, which continued to secrete high levels of IL-6 (Data S3). Amikacin was also added into uninfected macrophage cultures as a control. Supernatants were subsequently filtered through a 0.2 µm syringe filter, concentrated using Amicon Ultra Centrifugal Filters (3 kDa MWCO; Merck Millipore, Darmstadt, Germany), and stored at –20°C until use.

## Intracellular growth study of MABS by colony-forming unit assays and confocal microscopy

To investigate intracellular growth, human dermal fibroblasts were infected with MABS at an MOI of 5:1 for 3 h. The bacterial inoculum was then removed, and cells were washed three times with medium. Extracellular bacteria were eliminated by treatment with amikacin (200 mg/L) for 90 min. Infected cells were subsequently maintained in DMEM supplemented with 12% FBS and amikacin (20 mg/L) to inhibit extracellular bacterial growth. CFU assays were performed on days 0, 1, 3, 5, 7, and 9 post-infection. Briefly, cells were washed twice to remove antibiotic-containing medium and then lysed with sterile water containing 0.1% Triton X-100 to release intracellular bacteria. The lysates were diluted in PBS with 0.1% Tween-80, passed repeatedly through a 26-gage needle to reduce clumping, and plated onto Middlebrook 7H10 agar. Plates were incubated at 37°C for 2 days, followed by 1 day at room temperature.

To confirm intracellular localization of MABS within fibroblasts, cells were seeded onto 12 mm glass coverslips and infected as described above. On day 5 and day 9 post-infection, cells were washed with PBS, fixed with 3% paraformaldehyde for 15 min, and permeabilized with 0.1% Triton X-100 in PBS for 10 min. Indirect immunofluorescence staining was then performed using FITC mouse anti-human lysosome-associated membrane protein 1 (LAMP-1/CD107a; clone H4A3, BioLegend, San Diego, CA) at a 1:20 dilution and pooled human sera from patients with MABS infection at a 1:200 dilution. After washing with PBS containing 0.1% Tween-20, cells were incubated with secondary antibodies (10 µg/mL), including Alexa Fluor 488-conjugated goat anti-mouse IgG (Invitrogen) and DyLight 550-conjugated anti-human IgG (Abcam). Nuclei were counterstained with DAPI. Coverslips were mounted onto glass slides using Mowiol mounting medium and stored at 4°C. Stained cells were imaged using a ZEISS LSM 800 confocal microscope (Carl Zeiss, Germany). Image acquisition and processing were performed using ZEN lite 3.3 (Blue Edition) software.

## Cytokine expression and secretion analysis by qRT-PCR and ELISA

Total RNA was isolated from macrophages and fibroblast cell lysates using the GenUP Total RNA Kit (Biotechrabbit, Hennigsdorf, Germany). Complementary DNA (cDNA) was synthesized from the extracted RNA using SuperScript III Reverse Transcriptase (Invitrogen, Thermo Fisher Scientific). Quantitative real-time RT-PCR (qRT-PCR) was performed using SYBR Green master mix on a CFX96 Real-Time PCR Detection System (Bio-Rad, Hercules, CA). Gene expression levels were normalized to GAPDH, and relative fold changes were calculated using the $2^{-\Delta\Delta Cq}$ method.

For cytokine quantification by ELISA, IL-6 and IL-10 levels were measured using ELISA kits from FineTest (Wuhan, China), while TNF levels were determined using ELISA kits from Abcam, following the manufacturers' instructions.

## B cell/plasmablast culture and cell stimulation experiments

Human B cells were isolated from PBMCs using the EasySep Human B Cell Isolation Kit (STEMCELL Technologies, Vancouver, Canada) and cultured as previously described (23), with slight modifications. Briefly, purified B cells were cultured in RPMI 1640 supplemented with 10% human AB serum, IL-2 (20 U/mL), IL-10 (40 ng/mL), IL-15 (10 ng/mL), and

IL-21 (25 ng/mL). All cytokines were obtained from ImmunoTools (Friesoythe, Germany). To induce B cell proliferation and differentiation into plasmablasts, the TLR7/8 ligand R848 (Cell Signaling Technology, Danvers, MA) was added to the culture at a final concentration of 1 µg/mL for the first 4 days. The culture medium was replaced every 2 days. On day 4, cells were centrifuged at $300 \times g$ for 5 min to remove residual R848. Although the percentages of T cells were very low on day 0, the population expanded during culture, and their proportion increased by day 4 (Data S4). To eliminate these contaminating cells, magnetic cell separation was performed. Briefly, cell pellets were resuspended in PBS containing 2 mM EDTA, 1% human AB serum, and 2% FBS and incubated with biotinylated anti-CD3 (clone OKT3, BioLegend) and anti-CD56 (clone LT56; Exbio, Vestec, Czech Republic) antibodies at 4°C for 15 min. Cells were then incubated with MojoSort streptavidin-conjugated magnetic nanobeads (BioLegend) for 5 min. The tube containing the cell suspension was passed through a magnetic separator to deplete CD3$^+$ T cells and CD56$^+$ NK cells. Early plasmablasts in the negative fraction were collected for subsequent experiments.

To assess the effects of IL-6 and cytokines secreted by MABS-infected macrophages, early plasmablasts (day 4) were cultured for an additional 3 days in RPMI 1640 supplemented with 10% human AB serum, IL-2, and IL-15, in the presence of one of the following stimuli: (i) IL-6 (10 ng/mL, ImmunoTools; standard condition), (ii) no additional cytokines, (iii) supernatant from infected macrophages diluted to contain IL-6 at 10 ng/mL, or (iv) equally diluted supernatant from uninfected macrophages. On day 7, late plasmablast markers were analyzed by flow cytometry.

To investigate the roles of IL-6 and the Janus kinase (JAK) signaling pathway in supporting plasmablast survival in cultures treated with macrophage supernatants, early plasmablasts (day 4) were cultured in the presence of one of the following reagents: (i) tocilizumab (a humanized anti-IL-6 receptor monoclonal antibody, Actemra, 20 µg/mL; Roche), (ii) human IgG1 isotype control (SouthernBiotech, Birmingham, AL), (iii) baricitinib (a JAK inhibitor, 100 nM; Chemscene, Monmouth Junction, NJ), or (iv) DMSO vehicle control.

To assess IgG secretion, late plasmablasts (day 7) were washed three times with RPMI 1640 to remove residual IgG derived from human serum, tocilizumab, or isotype control. The cells were then cultured for an additional day in RPMI 1640 supplemented with 10% FBS, IL-10 (20 ng/mL), BAFF (100 ng/mL), and IFN-β (50 ng/mL; ProSpec Bio, Rehovot, Israel). On day 8, culture supernatants were collected, and IgG levels were quantified by ELISA using reagents from Mabtech (Stockholm, Sweden).

## Flow cytometric analysis of plasmablast markers

Cultured plasmablasts were incubated with Ghost Dye Red 710 fixable viability dye (Cell Signaling Technology) in PBS to exclude dead cells, followed by staining with fluorochrome-conjugated antibodies in flow cytometry staining buffer (PBS supplemented with 2% FBS and 1% human AB serum to block Fc receptors). The antibodies used were Elab Fluor Violet 450 anti-CD19 (clone CB19), Alexa Fluor 488 anti-CD27 (clone O323; Elabscience, Wuhan, China), BV510 anti-CD20 (clone 2H7), PE/Dazzle 594 anti-CD138 (clone MI15), APC/Fire 750 anti-CD3 (clone UCHT1; BioLegend), and PE/Dy747 anti-CD38 (clone HIT2; ImmunoTools). Cells were incubated with antibodies at 4°C for 15 min, washed with staining buffer, and fixed with 3% paraformaldehyde for 15 min. After a final wash, cells were resuspended in staining buffer for analysis. Unstained cells and single-stained compensation beads were used as controls. Fluorescence minus one (FMO) controls were included for CD38 and CD138. Data were acquired using a BD LSRFortessa flow cytometer (BD Biosciences) and analyzed with FlowJo software.

## Statistical analyses

Statistical analyses were performed using GraphPad Prism version 10.3.0 (GraphPad Software, San Diego, CA). Depending on the experimental design, unpaired Student's *t*-test, paired *t*-test, or one-way ANOVA followed by Tukey's post-hoc test was used to

assess statistical significance. A *P*-value of less than 0.05 was considered statistically significant.

## RESULTS

### MABS-infected macrophages produce high levels of cytokines, particularly IL-6

Macrophages infected with MABS expressed pro-inflammatory cytokine transcripts, including *TNF*, *IL-6*, *IL-12/23p40*, and *IL-23p19*, with expression levels remaining elevated at both 6 and 24 h post-infection. In contrast, gene expression levels in Pam3Cys-stimulated macrophages were higher at 6 h than at 24 h post-infection. We also examined the gene expression levels of cytokines and growth factors important for B cell activation, including IL-10, BAFF, and APRIL. We found that IL-10 mRNA expression was slightly upregulated at 6 h post-infection, whereas BAFF and APRIL expression remained unchanged or were slightly downregulated (Fig. 1A).

We further confirmed the secretion of TNF, IL-6, and IL-10 by ELISA and found that MABS-infected macrophages secreted high levels of IL-6 (>8 ng/mL) at 24 h post-infection. In contrast, the levels of secreted TNF and IL-10 were much lower than those of IL-6 (Fig. 1B). Since IL-6 is a well-known innate cytokine important for B cell activation, we hypothesized that high levels of IL-6 may enhance plasmablast survival and antibody production during MABS infection.

### MABS survives in human dermal fibroblasts and induces high levels of IL-6 production

Since fibroblasts are non-immune cells capable of secreting high levels of IL-6 (24, 25), we investigated whether MABS can invade, survive within human dermal fibroblasts, and induce IL-6 secretion. Using a colony-forming unit (CFU) assay, we observed a slight decrease in the number of MABS from day 0 to day 1, suggesting that some extracellular bacteria may have remained strongly adherent to the cell surface during the initial phase. Subsequently, intracellular MABS increased from day 1 to day 5 and became more pronounced from day 5 to day 9 (Fig. 2A), indicating successful bacterial survival and replication within the fibroblasts. Confocal microscopy analysis revealed that only a small proportion of fibroblasts were infected, with each infected cell typically harboring bacteria either individually or in clusters. Intracellular localization of MABS was confirmed by its colocalization with lysosome-associated membrane protein 1 (LAMP-1) (Fig. 2B and Data S5).

We also examined cytokine production by fibroblasts following infection. The results showed upregulation of *IL-6* and *TNF* transcripts, while *IL-10* mRNA was undetectable (Fig. 2C). However, ELISA analysis detected secretion of IL-6 only, with no detectable TNF protein. Notably, even uninfected fibroblasts secreted high basal levels of IL-6, which increased markedly upon infection (Fig. 2D).

### Supernatants from MABS-infected macrophages enhance plasmablast survival

Since IL-6 is well known to play a key role in the survival of plasmablasts or plasma cells (13, 14), we investigated whether IL-6-rich supernatants from macrophages can promote plasmablast survival during *in vitro* culture. For plasmablast generation, human B cells were isolated from PBMCs and stimulated with R848 (a TLR7/8 ligand) for 4 days to induce differentiation into early plasmablasts. Subsequently, the cultures were maintained for an additional 3 days under the following conditions: without recombinant IL-6 (control), with IL-6 (standard protocol), or with supernatants from either MABS-infected or uninfected macrophages (Fig. 3A). Gating strategy for live plasmablasts was shown in Fig. 3B. Flow cytometry analysis on day 7 revealed that the percentage of viable late plasmablasts was significantly higher in conditions containing IL-6 compared to those without IL-6. Similarly, cultures treated with infected macrophage supernatants exhibited

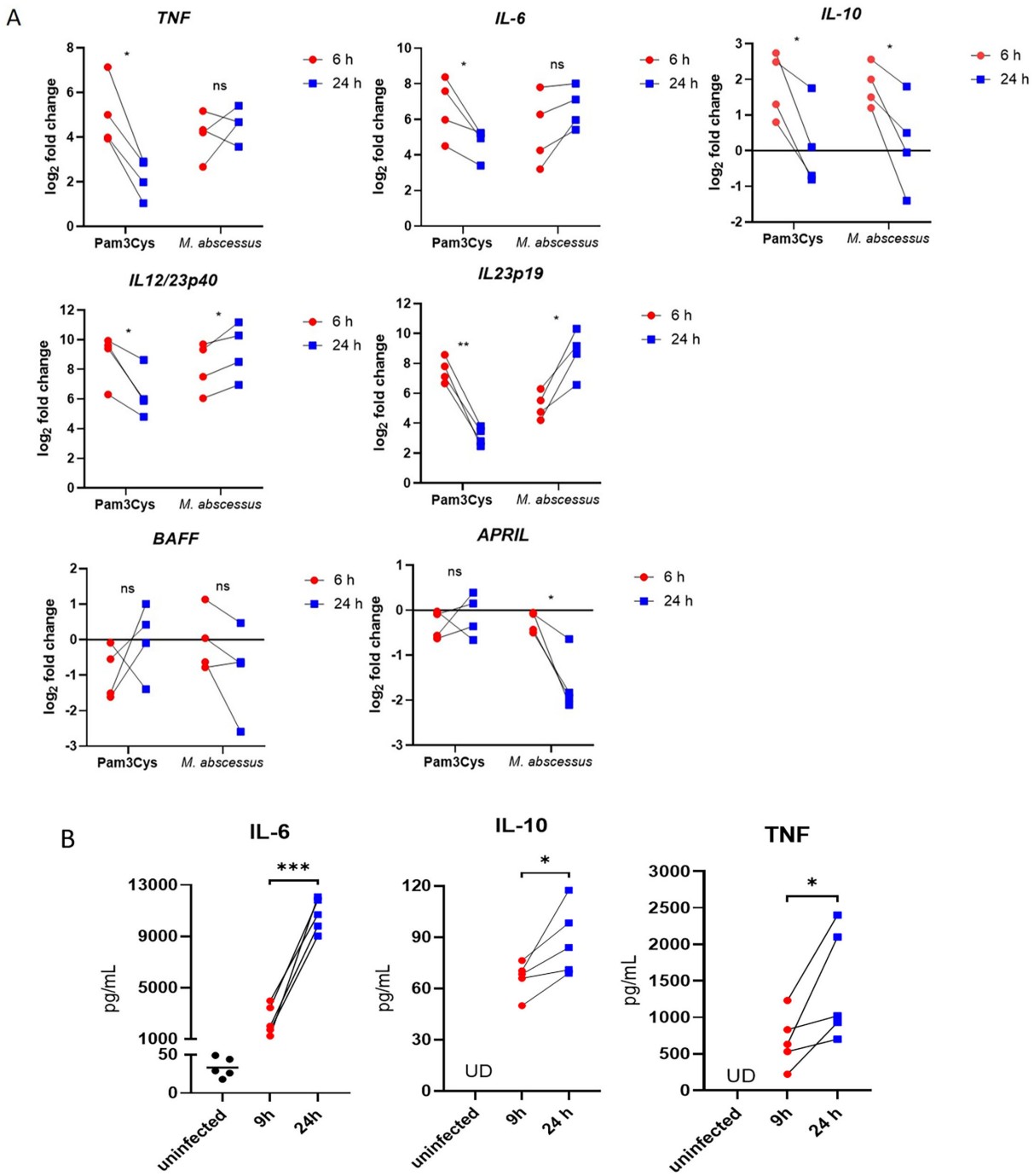

**FIG 1** Cytokine production in MABS-infected macrophages. (A) Macrophages were infected with MABS at an MOI of 20:1 or stimulated with Pam3Cys (a TLR2 ligand, 1 µg/mL) for 6 or 24 h. Cytokine mRNA levels were analyzed by qRT-PCR and are presented as $\log_2$ fold changes relative to unstimulated cells. (B) ELISA results showing cytokine levels in uninfected and MABS-infected macrophages at 9 and 24 h. Data are from four (A) or five (B) independent donors. *$P < 0.05$, **$P < 0.01$, ***$P < 0.001$ by paired $t$-test; ns, not significant.

a higher percentage of viable late plasmablasts than those treated with uninfected macrophage supernatants (Fig. 3C), as evidenced by a higher percentage of live cells and an increased frequency of CD27$^{hi}$CD20$^{lo}$CD38$^{hi}$ plasmablasts (Data S6A). Plasmablast cultures supplemented with supernatants from infected macrophages without amikacin treatment showed similar results (Data S6B). These findings suggest that cytokines secreted by MABS-infected macrophages promote plasmablast survival and expansion, potentially by influencing plasmablast differentiation and/or proliferation. However, we

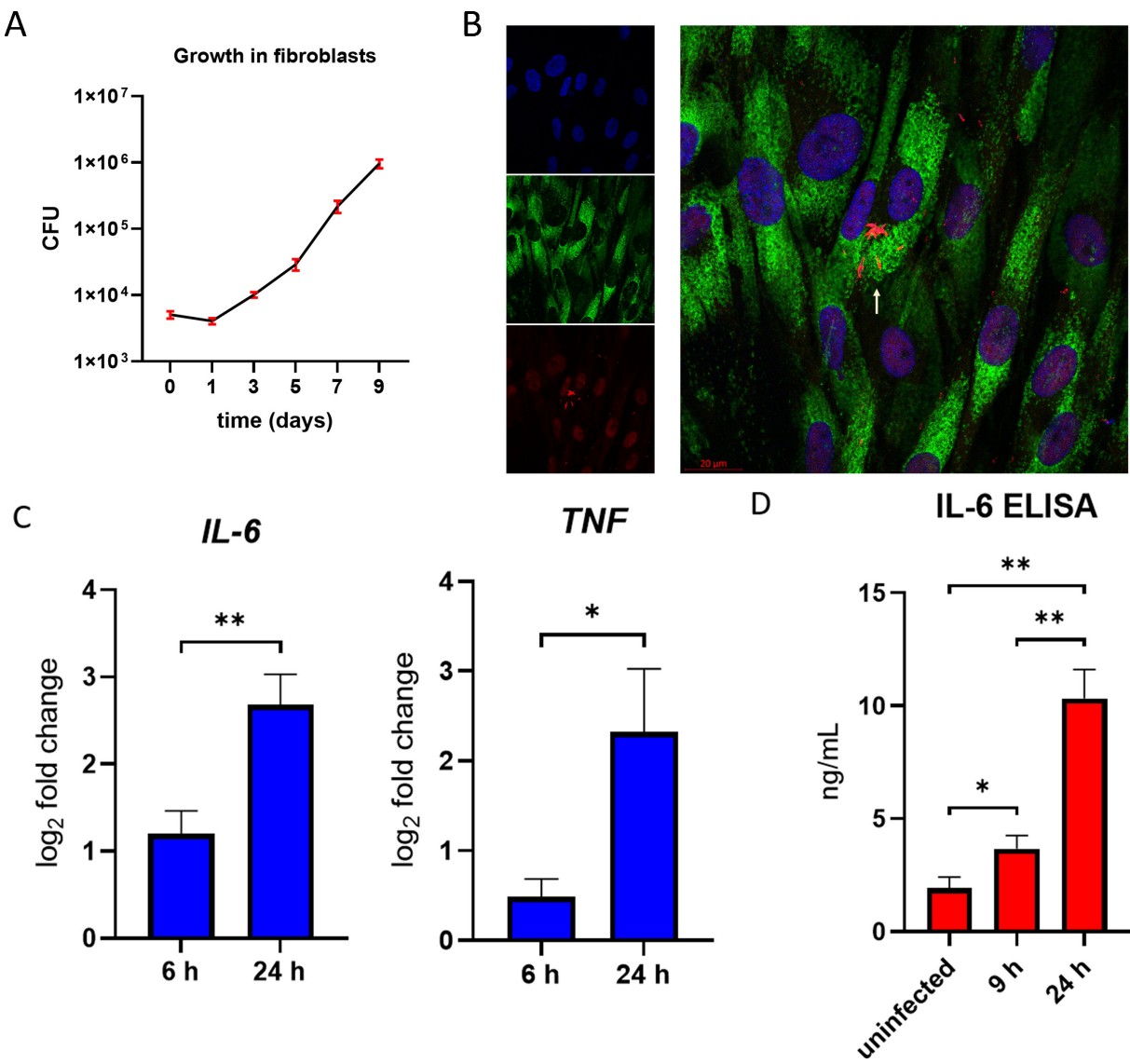

**FIG 2** Intracellular survival of MABS and cytokine production in human dermal fibroblasts. (A) Human dermal fibroblasts were infected with MABS at a multiplicity of infection (MOI) of 5:1 for 3 h. Cells were then washed and incubated with medium containing amikacin (200 µg/mL) for 90 min to eliminate extracellular bacteria. Infected cells were subsequently maintained in medium containing amikacin (20 µg/mL) to inhibit extracellular bacterial growth. Colony-forming unit (CFU) assays were performed on days 0, 1, 3, 5, 7, and 9. (B) Intracellular colocalization of MABS (red) and LAMP-1 (green) on day 9. Cell nuclei (blue) were counterstained with DAPI. (630 × magnification). (C) Fibroblasts were either left unstimulated or infected with MABS at an MOI of 20:1 for 6 or 24 h. Cytokine mRNA levels were analyzed by qRT-PCR and are presented as $\log_2$ fold changes relative to unstimulated cells. *$P < 0.05$ by unpaired $t$-test. (D) ELISA results showing cytokine levels in uninfected and MABS-infected fibroblasts at 9 and 24 h. *$P < 0.05$, **$P < 0.01$ by ANOVA with Tukey's post-hoc test. Data are presented as mean ± SD from three independent experiments.

did not compare the percentages of CD138[+] plasma cells, as CD138 expression on day 7 remained low and variable across donors (Data S4).

## Anti-IL-6R antibody (tocilizumab) reduces plasmablast survival and antibody secretion in cultures stimulated with supernatants from infected macrophages

We investigated whether IL-6 derived from infected macrophages is a key cytokine promoting plasmablast survival and expansion. To test this, we cultured early plasmablasts on day 4 with supernatants from MABS-infected macrophages and added either an anti-IL-6 receptor antibody (tocilizumab) or an isotype control. Binding of tocilizumab

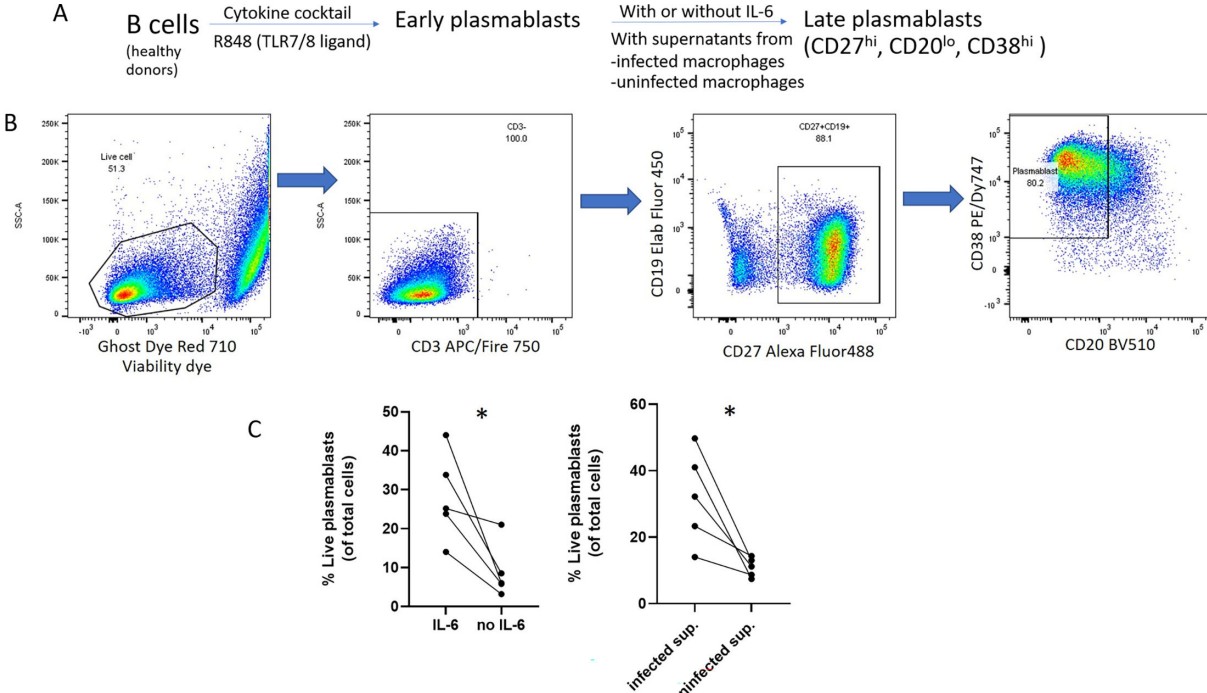

**FIG 3** Effect of IL-6 and infected macrophage supernatants on plasmablast survival. (A) Schematic diagram illustrating the plasmablast culture protocol. (B) Cells were differentiated into early plasmablasts using R848 for 4 days. Cells were then cultured for an additional 3 days in medium containing either no IL-6, IL-6 (10 ng/mL), infected macrophage supernatants adjusted to contain 10 ng/mL IL-6, or uninfected supernatants. (B) Gating strategy for live plasmablasts. Live cells were identified by exclusion of dead cells using a fixable viability dye (Ghost Dye Red 710). From the live cell population, CD3$^+$ T cells were excluded (CD3$^-$), followed by selection of CD27$^+$ cells. Plasmablasts were then gated based on low CD20 and high CD38 expression. (C) The percentage of live plasmablasts (of total cells) was calculated using the following formula: % live plasmablasts = % live cells × % CD3$^-$ cells × CD27$^{hi}$ cells × CD38$^{hi}$ CD20$^{lo}$ cells. Data are from five independent donors. *$P$-value < 0.05 by paired $t$-test comparing conditions with and without IL-6 or between infected and uninfected macrophage supernatants.

to the IL-6 receptor and its ability to inhibit STAT3 phosphorylation were confirmed by flow cytometry (Data S7A and B). Tocilizumab treatment reduced the percentage of viable late plasmablasts in a dose-dependent manner, with a detectable effect at 10 µg/mL and a more pronounced inhibition at 20 µg/mL (Data S7C; Fig. 4A). We also tested the effect of baricitinib, a clinically approved Janus kinase (JAK) inhibitor. Bariciti-nib was selected because of its ability to inhibit JAK1 and JAK2, which mediate IL-6 receptor signaling and act upstream of STAT3 activation (26). Although baricitinib (100 nM) was confirmed to inhibit IL-6-mediated STAT3 phosphorylation (Data S7D), it had no effect on plasmablast survival in contrast to tocilizumab (Fig. 4A). Since TNF was also secreted by macrophages at high levels, we tested the effect of TNF neutralization and found that anti-TNF treatment did not affect the percentage of live plasmablasts (Data S7E). Finally, the inhibitory effect of tocilizumab was further supported by a reduction in IgG secretion (Fig. 4B). Collectively, these findings suggest that IL-6 produced by MABS-infected macrophages enhances plasmablast survival and expansion and promotes antibody secretion, highlighting IL-6 as a potential target for further investigation in regulating antibody responses during infectious diseases.

## DISCUSSION

MABS is a common opportunistic pathogen in patients with adult-onset primary immunodeficiency associated with anti-IFN-γ autoantibodies. We hypothesized that cytokines released from MABS-infected cells may promote plasmablast survival and antibody production. Our findings demonstrate that MABS infection induces high levels of IL-6 secretion in both macrophages and fibroblasts. Notably, MABS was also able to survive within fibroblasts, indicating that, in addition to macrophages, non-immune cells

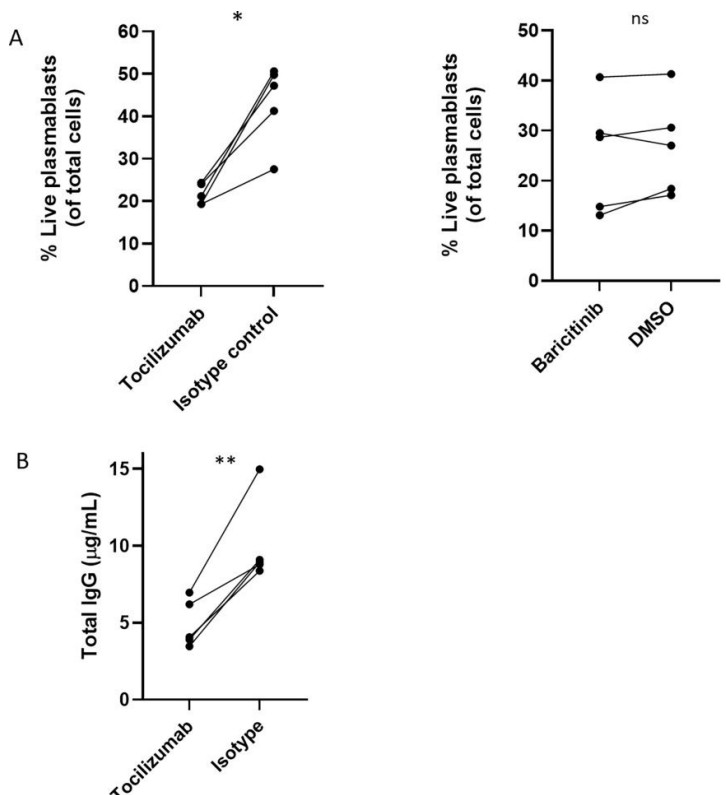

**FIG 4** Effects of tocilizumab and baricitinib on plasmablast survival (A) and IgG secretion (B). On day 4, plasmablast cultures were treated with one of the following conditions: (i) tocilizumab (20 µg/mL), (ii) isotype control antibody, (iii) baricitinib (100 nM), or (iv) DMSO vehicle control. Cultures were then stimulated with supernatants from MABS-infected macrophages for 3 additional days. On day 7, the percentage of plasmablasts was assessed by flow cytometry. IgG secretion was measured on day 8 following removal of residual IgG by washing out the old culture media. *$P < 0.05$, **$P < 0.01$, ns (not significant).

such as fibroblasts can serve as reservoirs for infection. We further demonstrated that IL-6 is a key cytokine produced by infected macrophages that promotes plasmablast survival and antibody secretion, as this effect was inhibited by tocilizumab, an anti-IL-6 receptor (IL-6R) blocker. However, the JAK inhibitor baricitinib did not affect plasmablast survival and expansion, which may be explained by its relatively high half-maximal inhibitory concentration ($IC_{50}$) for STAT3 (20–100 nM) compared to its $IC_{50}$ for JAK1/2 (~5 nM) (27). Although we demonstrated that baricitinib inhibits the early phase of IL-6-mediated STAT3 phosphorylation, it remains to be determined whether baricitinib modulates the late and long-term effects of IL-6. Collectively, our findings suggest that elevated IL-6 levels during active infection may contribute to dysregulated antibody responses. Therefore, effective microbial control is essential to prevent excessive IL-6 production. Moreover, IL-6 represents a potential therapeutic target warranting further clinical investigation.

Although macrophages are widely recognized as the primary host cells targeted by mycobacteria, several studies have shown that MABS is also capable of invading certain non-immune cells, including respiratory epithelial cells (28–30) and fibroblasts (31, 32). Our findings support these earlier observations by confirming the intracellular survival of MABS within dermal fibroblasts. While the host response of non-immune cells to MABS infection remains poorly characterized, a previous study reported no upregulation of IL-6 gene expression in infected respiratory epithelial cells (33). In contrast, our study demonstrates that dermal fibroblasts not only produce high basal levels of

IL-6 but also significantly increase IL-6 secretion upon infection. Since lymphoid tissues contain a specialized stromal cell type known as fibroblastic reticular cells (34), future studies employing these stromal cells are warranted to elucidate their potential role in modulating antibody response during mycobacterial infection.

Our results suggest that cytokines, particularly IL-6, produced during active infection may promote plasmablast survival and antibody production. Therefore, initiating effective antimicrobial therapy as early as possible is crucial. However, controlling MABS infection remains challenging due to its extensive drug resistance. Many clinical isolates of MABS exhibit resistance to conventional antibiotics, including macrolides and β-lactams such as imipenem and cefoxitin. Tigecycline, along with amikacin, is currently among the most active antimicrobial agents (35, 36) and is recommended during the initial phase of empirical treatment in many high-income countries (37, 38). Unfortunately, tigecycline remains expensive and unaffordable for most patients in resource-limited countries. We, therefore, hypothesize that the lack of access to tigecycline in empirical antimicrobial regimens may result in inadequate microbial control, sustained inflammation, and prolonged autoantibody production in patients from resource-limited settings.

Clinical data from patients with autoimmune diseases demonstrate that IL-6, even at concentrations as low as 1 ng/mL, enhances plasmablast survival and autoantibody secretion (39). Consistently, anti-IL-6R blockade reduces plasmablast survival (39) and autoantibody levels (40, 41). Tocilizumab has also been reported to decrease IL-21 production by T cells (40). Given that IL-21 is a signature cytokine of follicular helper T cells (42), IL-6 blockade may impair T cell-dependent B cell activation in addition to its role in disrupting IL-6-mediated plasmablast survival. Consistent with these findings, our study underscores the importance of macrophage-derived IL-6 in promoting plasmablast survival and suggests that targeting IL-6 with tocilizumab warrants further clinical investigation.

Our results demonstrate that tocilizumab did not completely inhibit plasmablast survival and expansion. These findings may be explained by several factors. First, the concentration of IL-6 used in our B-cell cultures was relatively high (10 ng/mL); therefore, tocilizumab may not have fully neutralized IL-6 activity under these conditions. Second, IL-6 is not the sole factor contributing to plasmablast survival. Indeed, other cytokines or chemokines from innate immune cells, such as IL-10, IL-12, and CXCL10, have been reported to promote plasmablast differentiation and survival (18, 43–45). Further studies are required to delineate the contributions of additional cytokines to plasmablast survival during MABS infection.

Although our *in vitro* experiments were conducted using B cells from healthy donors and measured total IgG secretion, we believe the findings may be relevant to pathogenic antibody production in patients. We attempted to perform similar cultures using B cells from patients; however, we were unable to obtain a high yield of B cells, and patient-derived B cells/plasmablasts showed poorer survival than those from healthy donors. This may be attributable to patient-related factors, including advanced age, underlying comorbidities, and a history of recurrent infections, which may compromise immune cell fitness. Additionally, due to technical limitations, it is difficult to accurately detect low levels of specific anti-IFN-γ autoantibodies. Despite these limitations, our findings may provide a rational basis for further clinical investigation of tocilizumab as a potential therapeutic approach.

Although IL-6 may have a protective role in certain infections, its function in mycobacterial infections remains controversial. In contrast, the protective roles of other cytokines—such as TNF, IL-1β, the IL-12/IFN-γ axis—as well as reactive nitrogen intermediates, are well established as key components of the host defense against mycobacterial infections (46, 47). Early studies have demonstrated that IL-6 plays a role in host defense against *Mycobacterium tuberculosis* infection, as intravenous infection with *M. tuberculosis* was lethal in IL-6-deficient mice (48). In models using low-dose aerosol infection, although IL-6 was shown to be essential for early IFN-γ production and

initial microbial control, it was dispensable for the development of long-term protective immunity against *M. tuberculosis* infection (49, 50). However, subsequent studies suggest that IL-6 may impair host defense by inhibiting macrophage responses to IFN-γ (51) and suppressing IFN-γ-induced autophagy (52). In clinical studies, unlike anti-TNF monoclonal antibodies, tocilizumab has not been associated with an increased risk of tuberculosis reactivation (53, 54). However, its use has been linked to a higher risk of certain pyogenic bacterial infections (55).

## Conclusion

In this study, we demonstrated that MABS induces high levels of inflammatory cytokines, particularly IL-6, in macrophages. We also found that MABS can invade and replicate within fibroblasts, resulting in a substantial increase in IL-6 production. Among the cytokines examined, IL-6 emerged as the key factor produced by macrophages that supports the survival of B cell-derived plasmablasts. This conclusion is supported by our finding that blockade of the IL-6 receptor with tocilizumab reduced plasmablast survival and IgG production in cultures supplemented with macrophage supernatants. Collectively, our findings suggest that IL-6 may play a critical role in sustaining antibody-producing plasmablasts and support further investigation of IL-6 as a potential target for modulating plasmablast survival and antibody production.

### ACKNOWLEDGMENTS

This project is financially supported by Mahidol University, Grant R016320007. Issara Prachongsai is partially supported by Siriraj Graduate Scholarship and Walailak University.

We thank Assist. Prof. Dr. Ladawan Khowawisetsut for some technical assistance in flow cytometry.

### AUTHOR AFFILIATIONS

[1]Graduate Program in Immunology, Faculty of Medicine Siriraj Hospital, Mahidol University, Bangkok, Thailand
[2]School of Allied Health Sciences, Walailak University, Thasala, Nakhon Si Thammarat, Thailand
[3]Department of Immunology, Faculty of Medicine Siriraj Hospital, Mahidol University, Bangkok, Thailand

### AUTHOR ORCIDs

Wiwit Tantibhedhyangkul  http://orcid.org/0000-0003-0943-9012

### AUTHOR CONTRIBUTIONS

Issara Prachongsai, Conceptualization, Data curation, Investigation, Methodology, Validation, Writing – original draft, Writing – review and editing | Rawipas Saisuwan, Data curation, Formal analysis, Investigation, Methodology, Software, Validation, Visualization, Writing – review and editing | Pattanan Smosorn, Data curation, Formal analysis, Investigation, Methodology, Validation, Writing – review and editing | Nuttawan Kassaket, Data curation, Methodology, Validation, Writing – review and editing | Wiwit Tantibhedhyangkul, Conceptualization, Data curation, Formal analysis, Funding acquisition, Investigation, Methodology, Project administration, Supervision, Validation, Visualization, Writing – original draft, Writing – review and editing

### ADDITIONAL FILES

The following material is available online.

## Supplemental Material

**Data S1 to S3 (Spectrum02520-25-s0001.pdf).** Monocyte and macrophage surface marker expression, macrophage cell death and cytokine gene expression after Mab infection, and IL-6 levels under different infection conditions.
**Data S4 (Spectrum02520-25-s0002.pdf).** Flow cytometry data.
**Data S5 (Spectrum02520-25-s0003.tiff).** Confocal microscopy images.
**Data S6 and S7 (Spectrum02520-25-s0004.pdf).** Plasmablasts on day 7 of culture treated with supernatants from infected or uninfected macrophages, and binding of tocilizumab to IL-6 receptor and its ability to inhibit STAT3 phosphorylation.
**Table S1 (Spectrum02520-25-s0005.pdf).** Primer sequences.

## Open Peer Review

**PEER REVIEW HISTORY (review-history.pdf).** An accounting of the reviewer comments and feedback.

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
