## [Reviewer comments · Microbiology Spectrum]

Microbiology Spectrum

Interleukin-6 from *Mycobacterium abscessus*-Infected Macrophages Enhances the Survival of B Cell-Derived Plasmablasts *In Vitro*

Issara Prachongsai, Rawipas Saisuwan, Pattanan Smosorn, Nuttawan Kassaket, and Wiwit Tantibhedhyangkul

Corresponding Author(s): Wiwit Tantibhedhyangkul, Mahidol University Faculty of Medicine Siriraj Hospital

Review Timeline:

Submission Date:	August 13, 2025
Editorial Decision:	September 15, 2025
Revision Received:	January 13, 2026
Editorial Decision:	January 27, 2026
Revision Received:	March 9, 2026
Accepted:	March 13, 2026

Editor: M. Victoria Delpino

Reviewer(s): Disclosure of reviewer identity is with reference to reviewer comments included in decision letter(s). The following individuals involved in review of your submission have agreed to reveal their identity: Peng GAO (Reviewer #1); Bethany Patenall (Reviewer #3)

Transaction Report:

DOI: <https://doi.org/10.1128/spectrum.02520-25>

Re: Spectrum02520-25 (**Interleukin-6 from *Mycobacterium abscessus*-Infected Macrophages Enhances the Survival of B Cell-Derived Plasmablasts *In Vitro***)

Dear Dr. Wiwit Tantibhedhyangkul:

Thank you for the privilege of reviewing your work. Below you will find my comments, instructions from the Spectrum editorial office, and the reviewer comments.

The overall conclusions of the manuscript appear sound; however, several controls require clarification or a clear justification for their use. The current data do not convincingly support the claim that these microbes replicate intracellularly at the time points tested. This point should be addressed by providing a statistical assessment of significance and/or by extending the incubation period-potentially to 7-9 days post-infection rather than 5-to allow bacterial numbers to reach meaningful levels.

Revision Guidelines

Sincerely,
M. Victoria Delpino
Editor
Microbiology Spectrum

Reviewer #1 (Comments for the Author):

Mycobacterium abscessus infections in patients with autoantibodies against interferon-gamma (IFN- γ) can cause severe

disseminated disease. This study investigated how the infection might promote these harmful autoantibodies. Researchers found that *M. abscessus*-infected macrophages and fibroblasts secrete high levels of the cytokine IL-6. Given IL-6's known role in supporting antibody-producing plasmablasts, its effect was tested on human B-cells differentiated into plasmablasts. Both recombinant IL-6 and supernatants from infected macrophages significantly enhanced plasmablast survival and IgG secretion. Crucially, blocking the IL-6 receptor with tocilizumab reversed this effect. These findings indicate that infection-induced IL-6 drives autoantibody persistence by promoting plasmablast survival, suggesting that targeting the IL-6 pathway could be a viable therapeutic strategy for these patients.

I have some comments:

Major:

1. The authors want to propose that enhanced IL-6 induce autoantibodies, but they just detect total IgG. Please try to confirm the autoantibody level.
2. They based on figure 1 to hypothesis that IL-6 may enhance autoantibody. There is no reason not consider the other two cytokines, TNF and IL-10. Even though their levels are low, they were originally undetectable. Please have an assay to compare their effect.
3. In figure 2B, the images from different field need to provide and full image and cropped image also need to provide. Indicate the color in images
4. The assay to prove that tocilizumab do have effect on IL-6 receptor and baricitinib do inhibit JAK is needed.
5. Based on the flow cytometry data in supp Data 4, compare tocilizumab at 20 ug/ml and 0, the % of live plasmablasts decreased nearly 8.2%, but total survival cell only decreased 4.9%. This means that other cells survival increased 3.3%. Does it mean that tocilizumab increased the survival of other cells?

Minor:

1. Primers should be provided
2. Not sure the format reason or not, the font size varied for the whole manuscript
3. line 247, it was comparing 6h vs 24h or comparing these two time point against 0h? if you want to say both elevated, that means comparing treated with untreated, why IL-6 group is ns?
4. Line 254, IL6 at 5-10 ng/ml, please double confirm the range
5. Figure 1B, please confirm the comparison groups
6. Figure 2, no need to mention this (undetectable data) in figures; Please add the bar to indicate which two groups you are comparing; Why use unpaired t-test, should be non-parametric test
7. Figure 4B, be consistent for the label
8. Line 324-315, there is no logic connection between the two sentences
9. Line 321-323, How to explain there is no effect of JAK inhibitor. It is not clear
10. Figure 4A, add control of uninduced cell culture. I know it can not compare with cells from different samples. But roughly, based on the data on figure 3C, the level will be under 20%. Here even Tocilizumab decreased the level, but still higher than 20%. Does it mean that IL-6 is not the only factors.

Reviewer #2 (Comments for the Author):

The authors present compelling data that suggests that cellular infection with *Mycobacterium abscessus* induces IL-6 that subsequently promotes plasmablast survival and may contribute to sustained autoantibody responses in patients. They note that targeting IL-6 signaling could represent a potential therapeutic strategy to reduce anti-IFN- γ autoantibody levels in infected individuals.

Overall, the manuscript is well written, and the conclusions drawn are interesting. The authors data demonstrating the *Mycobacterium abscessus* induces IL-6 that subsequently promotes plasmablast survival is strong. The data they present demonstrating the intracellular growth of *Mycobacterium abscessus* is less so. Additional statistical confirmation and/or additional experimentation is needed to make this conclusion.

Major Comments:

Lines 138-141: This appears to be counter-intuitive to the author's main conclusion(s) from the study. It stands to reason that plasmablasts cannot produce anti-IFN- γ autoantibodies in human patients if they are killed by the pathogen ... perhaps if lower MOIs were used for infections, the use of amikacin would not be necessary? Have the authors checked to see if co-incubation in the absence of amikacin resulted in cytotoxic effects? Have they considered that amikacin-treatment might be enhancing the stimulatory expression of cytokines from *Mycobacterium*-infected macrophages above that which would normally be present under native infection conditions?

Lines 265-267: Please provide statistics on these comparisons. From the data presented, it is questionable as to whether these microbes are replicating over the time span examined.

Lines 267-268: This low level of infection is not too surprising, given the MOIs used and the static (rather than rocking or centrifugal) infection conditions.

Figure 2B: This image is great for demonstrating the relative abundance of M.a. in the fibroblast population, but it would be nice to be able to observe the images in each channel in addition to the overlay provided.

Minor Comments:

Lines 131-132: Can the authors please add a justification for why this particular MOI was used and how well it corresponds to the host-microbial interactions that occur during the course of human infections?

Lines 135-136: Readers may benefit from a justification for why TLR1/2 agonists are the most appropriate stimulatory control for this experiment, given what is known about the primary PAMP(s) associated with Mycobacterium abscessus infections.

Lines 148-149: The MOI used here is 1/4th that of the cytokine assay. A justification for the change would appear warranted.

Lines 150-151: Did the authors assess impacts on host cell mitochondria associated with this elevated dose of amikacin? There is somewhat less of a concern if uninfected cells were treated in a similar fashion prior to collection of their cell supernatants

Lines 153-157: Were infected cells washed prior to lysis? If not, there is some concern that residual amikacin might be impacting subsequent colony counts.

Lines 171: Please include data on how many images / imaging fields were conducted per experimental group.

Line 208: 'equally diluted supernatant from uninfected macrophages' - were these uninfected macrophages subjected to prolonged amikacin treatment (like infected macrophages)?

Reviewer #3 (Comments for the Author):

Define abbreviations of bacterial names during first use and use abbreviations there after

Define TLR

Explain why reducing anti IFN gamma autoantibody levels would be advantageous

Explain clinical problem more clearly

Introduction:

More correct to say that mycobacterium in general are difficult to treat owing to resistance to multiple classes of antibiotics, this problem is not specific to M. Abscesses

Is colony morphology relevant to this manuscript?

Could Adult-onset immunodeficiency be abbreviated to increase readability of manuscript

You saw that M. Abscessus is most common infection in these patients, this needs a reference and it would be advantageous to quote a figure for clinical relevance

Methods:

Specific that strain was purchased from American Tissue Culture Collection

What media are cultures cryopreserved in (specify glycerol percentage)

You frequently highlight that this is common in asian patients, was the healthy donor blood demographically relevant?

Very thorough methods would these benefit being in supplementary material?

Results:

For CFU/fibroblast experiment was there a control to ensure no decrease in CFU was observed in just media?

Discussion:

The conclusion that M. Abscesses can survive in fibroblasts is quite substantial based only on the confocal imagery? Consider

changing to could serve as a a reservoir for infection and maybe suggest how this could be investigated?

Mention availability of tigecycline in Thailand, is this manuscript only focusing on Thai populations? It is confusing why Thailand specifically is mentioned rather than asian broadly

Comment on why B cell yield not high enough in patient populations

Response to reviewers

The overall conclusions of the manuscript appear sound; however, several controls require clarification or a clear justification for their use. The current data do not convincingly support the claim that these microbes replicate intracellularly at the time points tested. This point should be addressed by providing a statistical assessment of significance and/or by extending the incubation period-potentially to 7-9 days post-infection rather than 5-to allow bacterial numbers to reach meaningful levels.

Response to editor

We repeated CFU assays by extending the incubation period to 9 days and also repeated confocal microscopy experiment. The results are shown in new Fig.2A, 2B, supplementary data 5.

Reviewer #1 (Comments for the Author):

Mycobacterium abscessus infections in patients with autoantibodies against interferon-gamma (IFN- γ) can cause severe disseminated disease. This study investigated how the infection might promote these harmful autoantibodies. Researchers found that *M. abscessus*-infected macrophages and fibroblasts secrete high levels of the cytokine IL-6. Given IL-6's known role in supporting antibody-producing plasmablasts, its effect was tested on human B-cells differentiated into plasmablasts. Both recombinant IL-6 and supernatants from infected macrophages significantly enhanced plasmablast survival and IgG secretion. Crucially, blocking the IL-6 receptor with tocilizumab reversed this effect. These findings indicate that infection-induced IL-6 drives autoantibody persistence by promoting plasmablast survival, suggesting that targeting the IL-6 pathway could be a viable therapeutic strategy for these patients.

I have some comments:

Major:

Comment 1

1. The authors want to propose that enhanced IL-6 induce autoantibodies, but they just detect total IgG. Please try to confirm the autoantibody level.

We explain the reasons that we cannot perform experiments using patient-derived B cells in the discussion part, line 402–408.

We attempted to perform similar cultures using B cells from patients; however, we were unable to obtain a high yield of B cells, and patient-derived B cells/plasmablasts showed poorer survival than those from healthy donors. This may be attributable to patient-related factors, including advanced age, underlying comorbidities, and a history of recurrent infections, which may

compromise immune cell fitness. Additionally, due to technical limitations, it is difficult to accurately detect low levels of specific anti-IFN- γ autoantibodies.

Indeed, we had submitted this manuscript to mSphere but was rejected by the editor without peer review for this reason. Therefore, we resubmitted it to this journal and was eligible for peer review by this journal.

Comment 2

2. They based on figure 1 to hypothesis that IL-6 may enhance autoantibody. There is no reason not consider the other two cytokines, TNF and IL-10. Even though their levels are low, they were originally undetectable. Please have an assay to compare their effect.

To address this comment, we performed additional experiments to test the effect of TNF on plasmablast survival. We found that neutralizing anti-TNF have no effect on plasmablast survival and differentiation (Supplementary Data 7E, result line 329–332).

For IL-10, we did not perform additional experiments because its levels were very low (< 100 pg/mL). In addition, there is currently no FDA-approved anti-IL-10 drugs for clinical use.

Comment 3

3. In figure 2B, the images from different fields need to provide and full image and cropped image also need to provide. Indicate the color in images

We have included new confocal microscopy images in Figure 2B showing single-channel images, as well as Supplementary Data 5, which presents images from multiple fields.

Comment 4

4. The assay to prove that tocilizumab do have effect on IL-6 receptor and baricitinib do inhibit JAK is needed.

We performed additional experiments to confirm that tocilizumab binds to the IL-6 receptor and inhibits IL-6–mediated STAT3 phosphorylation (Supplementary Data 7A, 7B). Moreover, we demonstrated that baricitinib (100 nM) inhibits IL-6–mediated STAT3 phosphorylation (Supplementary Data 7D). These findings imply that baricitinib inhibits JAK1/2, which function upstream of STAT phosphorylation.

These results are described in line 320–328.

Comment 5

5. Based on the flow cytometry data in supp Data 4, compare tocilizumab at 20 ug/ml and 0, the % of live plasmablasts decreased nearly 8.2%, but total survival cell only decreased 4.9%. This means that other cells survival increased 3.3%. Does it mean that tocilizumab increased the survival of other cells?

We hypothesize that TCZ inhibits not only cell survival but also the expansion of plasmablasts.

As shown in the images, the percentage of CD27⁺ cells (plasmablasts) was higher in conditions without tocilizumab than in the presence of tocilizumab (20 µg/mL) (Supplementary Data 7C).

In addition, Supplementary Data 6A also shows that supernatants from infected macrophages increased both the percentage of live cells and the frequency of CD27^{hi}CD20^{lo}CD38^{hi} cells.

These findings suggest that IL-6 promote cell survival and plasmablast expansion, potentially by influencing plasmablast differentiation or proliferation.

To better support the conclusion that IL-6 promotes plasmablast survival and expansion and that this effect is prevented by tocilizumab, we revised and clarified the relevant sentences in the Results section, line 310–312, 333–335

Without tocilizumab

With tocilizumab 20 µg/mL

Minor:

Comment 1

1. Primers should be provided

We show the primer sequences in a Supplementary Table.

Comment 2

2. Not sure the format reason or not, the font size varied for the whole manuscript

We believe that the errors may have occurred during the submission process. Although the text file uses Times New Roman, 12-point font, the fonts used in the images may differ.

Comment 3

3. line 247, it was comparing 6h vs 24h or comparing these two time point against 0h? if you want to say both elevated, that means comparing treated with untreated, why IL-6 group is ns?

We apologize for the lack of clarity. Gene expression levels at 6 and 24 hours were first normalized to the 0-hour time point (unstimulated cells), which was set as the baseline or zero. Fold changes at 6 and 24 hours therefore represent induction relative to unstimulated cells.

We then compared the fold changes at 6 and 24 hours to each other to assess temporal differences in gene expression. Using this approach, IL-6 expression at 24 hours was not statistically different from that at 6 hours, resulting in a non-significant (ns) comparison between these two time points.

This finding is consistent with known cytokine expression kinetics, as TNF and IL-6 are early-response cytokines that typically peak between 4–8 hours and may either decline or remain sustained at later time points, such as 24 hours.

Importantly, at both 6 and 24 hours, the fold changes of several cytokines were clearly above zero, indicating upregulation relative to unstimulated cells. The absence of statistical significance for IL-6 reflects the comparison between 6 and 24 hours, not a lack of induction compared with baseline.

Comment 4

4. Line 254, IL6 at 5-10 ng/ml, please double confirm the range

We further confirmed the secretion of IL-6 by ELISA and found that *M. abscessus*-infected macrophages secreted high levels of IL-6 (>8 ng/mL) at 24 hours post- infection (line 271).

Comment 5

5. Figure 1B, please confirm the comparison groups.

We have edited this figure to improve the clarity of the statistical comparisons. We compared the 9 h and 24 h time points. Time 0 was not included in the comparison because the levels were very low or undetectable.

Comment 6

6. Figure 2, no need to mention this (undetectable data) in figures; Please add the bar to indicate which two groups you are comparing; Why use unpaired t-test, should be non-parametric test?

“Undetectable data” were removed from the figure.

We have added bars to indicate the comparison groups.

In Figure 2, the data were derived from infections in a fibroblast cell line. Because data from cell lines are generally normally distributed, we used parametric tests: an unpaired *t*-test for Figure 2C and ANOVA with Tukey’s post hoc test for Figure 2D.

Comment 7

7. Figure 4B, be consistent for the label.

We have used the symbol “ μ ” for micro in both figure and figure legends.

Comment 8

8. Line 324-325, there is no logic connection between the two sentences

We have modified the sentences as follows to improve clarity and logical flow: Line 353–356

Therefore, effective microbial control is essential to prevent excessive IL-6 production in immunodeficient patients with anti-IFN- γ autoantibodies. Moreover, IL-6 represents a potential therapeutic target that warrants further clinical investigation.

Comment 9

9. Line 321-323, How to explain there is no effect of JAK inhibitor. It is not clear.

In the revised manuscript, we demonstrate that baricitinib inhibits the early phase (20 min) of IL-6-mediated STAT3 phosphorylation (Supplementary Data 7D); however, it remains to be determined whether baricitinib negatively modulates the late and long-term effects of IL-6.

(Discussion line 349–352)

Comment 10

10. Figure 4A, add control of uninduced cell culture. I know it can not compare with cells from different samples. But roughly, based on the data on figure 3C, the level will be under 20%. Here even Tocilizumab decreased the level, but still higher than 20%. Does it mean that IL-6 is not the only factors.

We think that this finding can be explained by: (Discussion new paragraph, line 391–399)

1. The level of IL-6 used for supplementation in our B-cell cultures was relatively high (10 ng/mL); therefore, tocilizumab may not have completely inhibited the effects of IL-6.

2. IL-6 is not the sole factor contributing to plasmablast survival. Indeed, other cytokines or chemokines from innate immune cells, such as IL-10, IL-12, and CXCL10, have been reported to promote plasmablast differentiation and survival

Reviewer #2 (Comments for the Author):

The authors present compelling data that suggests that cellular infection with *Mycobacterium abscessus* induces IL-6 that subsequently promotes plasmablast survival and may contribute to sustained autoantibody responses in patients. They note that targeting IL-6 signaling could represent a potential therapeutic strategy to reduce anti-IFN- γ autoantibody levels in infected individuals.

Overall, the manuscript is well written, and the conclusions drawn are interesting. The authors data demonstrating the *Mycobacterium abscessus* induces IL-6 that subsequently promotes plasmablast survival is strong. The data they present demonstrating the intracellular growth of *Mycobacterium abscessus* is less so. Additional statistical confirmation and/or additional experimentation is needed to make this conclusion.

Major Comments:

Comment 1

Lines 138-141: This appears to be counter-intuitive to the author's main conclusion(s) from the study.

It stands to reason that plasmablasts cannot produce anti-IFN- γ autoantibodies in human patients if they are killed by the pathogen ... perhaps if lower MOIs were used for infections, the use of amikacin would not be necessary?

Have the authors checked to see if co-incubation in the absence of amikacin resulted in cytotoxic effects?

To address your concerns, we performed additional experiments in the revised manuscript and demonstrated that supernatants from infected macrophages without amikacin treatment also promoted plasmablast survival and expansion (Supplementary Data 6B). These effects were comparable to those observed with supernatants from amikacin-treated cultures (Result line 308–310).

Moreover, supernatants from uninfected macrophages also contained amikacin (Method line 159)

In the previous draft of the manuscript, amikacin was added to inhibit bacterial protein synthesis, as mycobacteria possess a type VII secretion system that transports effector proteins, some of which may exert cytotoxic effects on mammalian host cells.

Have they considered that amikacin-treatment might be enhancing the stimulatory expression of cytokines from *Mycobacterium*-infected macrophages above that which would normally be present under native infection conditions?

We do not think that low concentrations of aminoglycosides affect cellular responses, as aminoglycosides (e.g., streptomycin, gentamicin, or amikacin) are commonly used antibiotics in cell culture. Aminoglycosides can enter intracellular compartments and accumulate in endosomes or phagosomes after prolonged incubation (>24 hours); however, they do not penetrate efficiently into the cytosol of macrophages and are therefore not effective against cytosolic bacteria.

See Ref.

Maurin M, Raoult D. Use of aminoglycosides in treatment of infections due to intracellular bacteria. *Antimicrob Agents Chemother.* 2001 Nov;45(11):2977-86.

The introduction part of this article: Menashe O, et al. Aminoglycosides affect intracellular *Salmonella enterica* serovars typhimurium and virchow. *Antimicrob Agents Chemother.* 2008 Mar;52(3):920-6.

Emerging evidence suggests that aminoglycosides can enter the cytosol, reach organelles such as the endoplasmic reticulum and mitochondria, and induce cytotoxicity. However, these mechanisms have primarily been described in specific cell types, including renal tubular epithelial cells and inner ear hair cells, and likely occur after prolonged exposure.

Indeed, the “gentamicin or amikacin protection assay” is a well-established protocol in microbiology to eliminate extracellular bacteria while preserving intracellular bacteria. To study intracellular growth of *M. abscessus*, amikacin is routinely added to cell culture to inhibit extracellular bacterial growth

In addition, amikacin was also added to the culture medium of uninfected cells as a control (Method line 159).

Comment 2

Lines 265-267: Please provide statistics on these comparisons. From the data presented, it is questionable as to whether these microbes are replicating over the time span examined.

As suggested by the editor, we extended the experimental duration to 7–9 days and repeated the experiments four times. The results are shown in the new Figure 2A. The growth of intracellular mycobacteria became more evident after day 5.

Comment 3

Lines 267-268: This low level of infection is not too surprising, given the MOIs used and the static (rather than rocking or centrifugal) infection conditions.

We agree with your comment and have revised the relevant sentences in the revised manuscript (Line 279–286).

We also performed new experiments and attached new Figure 2A-B that show the growth of mycobacteria in fibroblasts and confocal microscopy image. In addition, we edited some sentences in result.

Comment 4

Figure 2B: This image is great for demonstrating the relative abundance of M.a. in the fibroblast population, but it would be nice to be able to observe the images in each channel in addition to the overlay provided.

Thank you for your comment. We attach new images in Figure 2B (showing images from each single channel) in this revised manuscript and add Supplementary Data 5 to show confocal images from different fields.

Minor Comments:

Comment 5

Lines 131-132: Can the authors please add a justification for why this particular MOI was used and how well it corresponds to the host-microbial interactions that occur during the course of human infections?

MOIs in the range of 10–20 are widely used in *M. abscessus* infection studies to ensure measurable bacterial uptake, intracellular localization, and host responses. Previous studies have employed a range of MOIs in infection experiments.

We performed preliminary experiments to optimize the MOI. When comparing MOIs of 10:1 and 20:1, cell death remained minimal at both conditions, whereas cytokine gene expression was slightly higher at an MOI of 20:1. In contrast, at an MOI of 40:1, cell death was markedly increased compared with the other two MOIs. Therefore, we selected an MOI of 20:1 for this study. The corresponding data are shown in the Supplementary Data 2, method line 146–147.

Comment 6

Lines 135-136: Readers may benefit from a justification for why TLR1/2 agonists are the most appropriate stimulatory control for this experiment, given what is known about the primary PAMP(s) associated with Mycobacterium abscessus infections.

Here is the edited sentence (Line 150 -152):

Pam3Cys-Ser-(Lys)₄ 1 µg/mL (Abcam, Cambridge, United Kingdom), a TLR1/TLR2 agonist, was used as a positive control for macrophage stimulation, as mycobacterial cell wall components are mainly recognized by TLR2.

(Hu W, Spaink HP. The Role of TLR2 in Infectious Diseases Caused by Mycobacteria: From Cell Biology to Therapeutic Target. *Biology (Basel)*. 2022 Feb 5;11(2):246.)

Comment 7

Lines 148-149: The MOI used here is 1/4th that of the cytokine assay. A justification for the change would appear warranted.

We used a lower MOI for the bacterial growth studies conducted over 9 days because we are afraid that a higher MOI could result in rapid host cell death. In contrast, a higher MOI was used for cytokine analyses performed over a short period (24 hours).

Comment 8

Lines 150-151: Did the authors assess impacts on host cell mitochondria associated with this elevated dose of amikacin? There is somewhat less of a concern if uninfected cells were treated in a similar fashion prior to collection of their cell supernatants

Incubation of infected cells with amikacin at 200 mg/L is a common protocol to kill extracellular *M. abscessus* (Ref. Zhu R et al., In vitro and intracellular inhibitory activities of nosiheptide against *Mycobacterium abscessus*. Front Microbiol. 2022 Jul 26;13:926361). A short incubation period is unlikely to enable amikacin to enter the cytoplasm of cells and affect mitochondria.

Comment 9

Lines 153-157: Were infected cells washed prior to lysis? If not, there is some concern that residual amikacin might be impacting subsequent colony counts.

Cells were washed twice to remove antibiotics prior to CFU assays (Line 171–172).

Comment 10

Lines 171: Please include data on how many images / imaging fields were conducted per experimental group.

We examined several microscopic fields (more than five) to identify intracellular bacteria, as the bacteria were difficult to detect. We did not describe this process in the Methods section because the number of fields scanned was not formally quantified. However, we present three representative images of infected cells for each time point (days 5 and 9 post-infection) in the Supplementary Data 5.

Comment 11

Line 208: 'equally diluted supernatant from uninfected macrophages' - were these uninfected macrophages subjected to prolonged amikacin treatment (like infected macrophages)?

Yes, the supernatants of uninfected cells also contain amikacin. (Method line 159)

Reviewer #3 (Comments for the Author):

Response to Reviewer #3

The authors thanks to the reviewer's comments. The manuscript has been revised accordingly as detailed below.

Comment 1:

Define abbreviations of bacterial names during first use and use abbreviations thereafter.

I have corrected as you suggested. We use 'MABS' as the abbreviation of *M. abscessus*.

Comment 2:

Define TLR.

We have now defined TLR as "Toll-like receptor (TLR)" at its first appearance in the Introduction and used TLR throughout the manuscript.

Comment 3:

Explain why reducing anti-IFN- γ autoantibody levels would be advantageous.

We explain in the introduction line...

Since IFN- γ is a critical cytokine for the type 1 immune response, these patients frequently experience recurrent opportunistic infections caused by intracellular pathogens (Line 64–66).

Decreases in these autoantibodies are associated with improved clinical outcomes of infection (Line 97–98).

Introduction

Comment 4:

Explain the clinical problem more clearly.

We add these sentences in the new paragraph of introduction (line 96 -105).

Moreover, in AOID patients, levels of anti-IFN- γ autoantibodies are increased during active infections.

Several immunosuppressive agents have been tried to reduce anti-IFN- γ autoantibody levels, as decreases in these autoantibodies are associated with improved clinical outcomes of infection.

Commonly used therapies include B cell–depleting agents (anti-CD20 monoclonal antibodies, such as rituximab) as well as non-specific immunosuppressive agents, including corticosteroids and cyclophosphamide. However, these immunosuppressive approaches have shown inconsistent or variable efficacy across studies (6, 19, 20). Moreover, non-specific immunosuppressive drugs adversely affect not only B cells but also T cells and phagocytes, potentially compromising host immunity to infection. Therefore, to identify novel therapeutic targets, we focus on macrophage-derived cytokines that may directly regulate antibody-secreting cells, including plasmablasts and plasma cells.

Comment 5:

It is more correct to say that *Mycobacterium* in general are difficult to treat owing to resistance to multiple classes of antibiotics; this problem is not specific to *M. abscessus*.

According to literature review, *Mycobacterium abscessus* is widely regarded as one of the most antibiotic-resistant or difficult-to-treat mycobacterial species.

Ref.

Johansen MD, Herrmann JL, Kremer L. Non-tuberculous mycobacteria and the rise of *Mycobacterium abscessus*. *Nat Rev Microbiol*. 2020 Jul;18(7):392-407.

Broncano-Lavado A, Senhaji-Kacha A, Santamaría-Corral G, Esteban J, García-Quintanilla M. Alternatives to Antibiotics against *Mycobacterium abscessus*. *Antibiotics (Basel)*. 2022 Sep 28;11(10):1322.

This can be explained by its susceptibility being largely limited to parenteral antibiotics, including amikacin (to which isolates are often susceptible but with high MICs of 8–16 mg/L, near the upper limit of the susceptibility breakpoint) and tigecycline (which is costly and may be unaffordable for some patients). In contrast, resistance to commonly used oral antibiotics, such as macrolides, is high.

Comment 6:

Is colony morphology relevant to this manuscript?

I think colony morphology is relevant. I use rough colony morphotype of *M. abscessus* throughout this manuscript because this morphotype is more virulent and more commonly found in patients.

Comment 7:

Could Adult-onset immunodeficiency be abbreviated to increase readability of the manuscript?

We have abbreviated “Adult-onset immunodeficiency” to “AOID” after its first full mention and used the abbreviation consistently throughout the text.

Comment 8:

You note that *M. abscessus* is the most common infection in these patients; this needs a reference and figure for clinical relevance.

We add the reference 4 and 21 in line 106–107.

Given that *Mycobacterium abscessus* is the most common opportunistic infection in immunodeficient patients with anti-IFN- γ autoantibodies (4, 21),...

Methods

Comment 9:

Specify that the strain was purchased from American Tissue Culture Collection.

We have clarified in the Methods section (Line 116) that *M. abscessus* (ATCC 19977) strain was obtained from the American Type Culture Collection (ATCC, Manassas, VA, USA).

Comment 10:

What media are cultures cryopreserved in (specify glycerol percentage)?

We have added the preservation condition in the Methods section as follows: “Bacterial stocks were cryopreserved in RPMI1640 with 10% (v/v) DMSO and stored at –80 °C.” (line 119 – 122)

Although glycerol is more commonly used for cryopreservation of mycobacteria, we observed that DMSO resulted in higher bacterial viability after thawing compared to glycerol.

Comment 11:

You frequently highlight that this is common in Asian patients. Was the healthy donor blood demographically relevant?

We don't think that the demographic data of healthy donors are relevant to the study. However, all of these healthy donors are Asians.

Comment 12:

Very thorough methods — would these benefit from being in supplementary material?

We believe that the detailed Methods section is important for readers to fully understand the experimental design and to ensure reproducibility of the study. However, we agree that the Methods can be further streamlined, and we will summarize and condense the section where appropriate prior to publication.

Results

Comment 13:

For CFU/fibroblast experiments, was there a control to ensure no decrease in CFU was observed in just media?

We performed a new CFU assay with prolonged incubation up to 9 days, which yielded clearer results than the previous assay. Based on the updated CFU data shown in Figure 2A, *M. abscessus* clearly replicates within fibroblasts. In addition, supernatants were collected to quantify extracellular bacteria, and the number of extracellular bacteria accounted for less than 10% of the intracellular bacterial load.

Discussion

Comment 14:

The conclusion that *M. abscessus* can survive in fibroblasts is quite substantial based only on the confocal imagery. Consider changing to “could serve as a reservoir for infection,” and suggest how this could be investigated.

The new Figure 2A shows that *M. abscessus* can replicate in fibroblasts. In addition, confocal images in the new Figure 2B demonstrate clusters of bacteria within fibroblasts. Together, these new figures provide stronger evidence for the intracellular replication of *M. abscessus* in fibroblasts and further support the Discussion and Conclusions.

Comment 15:

Mention availability of tigecycline in Thailand — is this manuscript only focusing on Thai populations? It is confusing why Thailand specifically is mentioned rather than Asia broadly.

We change this sentence:

Original version “Unfortunately, in Thailand, only the original brand of tigecycline (Tygacil®) is currently available, and its high cost makes it unaffordable for most patients.” was replaced with

“Unfortunately, tigecycline is still expensive and unaffordable for most patients in resource-limited countries.” Line 376 – 377.

Comment 16:

Comment on why B-cell yield was not high enough in the patient population.

Since the approval of ethic committee allows to give the consent for blood collection at 20 mL per participants. The limited blood volume affects the yield of B-cells in the experiment because B cells usually account for 5-10% of blood lymphocytes. In addition, the patients are mostly old and have recurrent infections and underlying conditions. We think that host health status may affect the survival of B cells in vitro culture.

We explain the reason in discussion line 402–408.

Re: Spectrum02520-25R1 (**Interleukin-6 from *Mycobacterium abscessus*-Infected Macrophages Enhances the Survival of B Cell-Derived Plasmablasts *In Vitro***)

Dear Dr. Wiwit Tantibhedhyangkul:

Thank you for the privilege of reviewing your work. Below you will find my comments, instructions from the Spectrum editorial office, and the reviewer comments.

Thank you for submitting your revised manuscript. The reviewers have now evaluated your revision; their comments are summarized below. I invite you to submit a further revised manuscript that addresses the points described. Reviewer 2 is satisfied with how you handled earlier concerns; Reviewer 1 raises a substantive issue that requires either additional experimental work or a substantial revision of the manuscript text and conclusions

Revision Guidelines

Sincerely,
M. Victoria Delpino
Editor
Microbiology Spectrum

Reviewer #1 (Comments for the Author):

For the first question, since you cannot accurately detect low levels of specific anti-IFN- γ autoantibodies, you can not make the

conclusion. Either you need to find a way to detect the autoantibodies or revise your conclusion or presentation related to autoantibody.

Reviewer #2 (Comments for the Author):

The authors have adequately addressed my prior concerns with potential indirect affects of their experimentation as well as characterization of the growth of the organism.

Response to reviewers (R2)

Reviewer 1

For the first question, since you cannot accurately detect low levels of specific anti-IFN- γ autoantibodies, you cannot make the conclusion. Either you need to find a way to detect the autoantibodies or revise your conclusion or presentation related to autoantibody.

We changed /removed the conclusion concerning related to autoantibodies as follows

1. contribute to the promotion of these autoantibodies.
To
Line 16
contribute to plasmablast survival and antibody production.
2. Collectively, these findings suggest that infection-induced IL-6 promotes plasmablast survival and may contribute to sustained autoantibody responses in patients
To
Line 30-31
Collectively, these findings suggest that infection-induced IL-6 promotes plasmablast survival and antibody production.
3. Targeting IL-6 signaling could therefore represent a potential therapeutic strategy to reduce anti-IFN- γ autoantibody levels.
To
Line 31-32
Targeting IL-6 signaling could therefore represent a potential therapeutic strategy to modulate antibody responses.
4. Together, these findings provide a rationale for future clinical trials investigation of cytokine-targeted approaches to reduce pathogenic autoantibody responses.
To
Line 46-47

Together, these findings provide a rationale for future clinical investigation of cytokine-targeted therapeutic approaches.

5. We further confirm that IL-6 plays a key role in promoting plasmablast survival and may represent a potential therapeutic target to modulate pathogenic autoantibody production.

To

Line 109-112

We further confirm that IL-6 plays a key role in promoting plasmablast survival and provide additional support for the role of myeloid cell-derived innate cytokines in regulating antibody production.

6. Collectively, these findings suggest that IL-6 produced by MABS-infected macrophages enhances plasmablast survival and expansion and may represent a therapeutic target in patients with aberrant autoantibody production.

To

Line 332-335

Collectively, these findings suggest that IL-6 produced by MABS-infected macrophages enhances plasmablast survival and expansion and promotes antibody secretion, highlighting IL-6 as a potential target for further investigation in regulating antibody responses during infectious diseases.

7. Collectively, our findings suggest that elevated IL-6 levels during active infection may contribute to aberrant autoantibody production. Therefore, effective microbial control is essential to prevent excessive IL-6 production in immunodeficient patients with anti-IFN- γ autoantibodies.

To

Line 351-353

Collectively, our findings suggest that elevated IL-6 levels during active infection may contribute to dysregulated antibody responses. Therefore, effective microbial control is essential to prevent excessive IL-6 production.

8. Consistent with these findings, our study underscores the importance of macrophage-derived IL-6 in promoting plasmablast survival and suggests that targeting IL-6 with tocilizumab may serve as a potential therapeutic option for patients with anti-IFN- γ autoantibody production.

To

Line 385-387

Consistent with these findings, our study underscores the importance of macrophage-derived IL-6 in promoting plasmablast survival and suggests that targeting IL-6 with tocilizumab warrants further clinical investigation.

9. Despite these limitations, we believe our findings provide a rational basis for further clinical investigation of tocilizumab as a potential therapeutic approach.

To

Line 405-406

Despite these limitations, our findings may provide a rational basis for further clinical investigation of tocilizumab as a potential therapeutic approach.

10. Collectively, our findings suggest that IL-6 may play a critical role in sustaining antibody-producing plasmablasts and support further investigation of IL-6 could represent a potential therapeutic target for reducing pathogenic autoantibody production.

To

Line 429-432

Collectively, our findings suggest that IL-6 may play a critical role in sustaining antibody-producing plasmablasts and support further investigation of IL-6 as a potential target for modulating plasmablast survival and antibody production.

Re: Spectrum02520-25R2 (**Interleukin-6 from *Mycobacterium abscessus*-Infected Macrophages Enhances the Survival of B Cell-Derived Plasmablasts *In Vitro***)

Dear Dr. Wiwit Tantibhedhyangkul:

Your manuscript has been accepted, and I am forwarding it to the ASM production staff for publication. Your paper will first be checked to make sure all elements meet the technical requirements. ASM staff will contact you if anything needs to be revised before copyediting and production can begin. Otherwise, you will be notified when your proofs are ready to be viewed.

Sincerely,
M. Victoria Delpino
Editor
Microbiology Spectrum